# Design Space Exploration for YOLO Neural Network Accelerator

**Hongmin Huang [1,2], Zihao Liu [1], Taosheng Chen [1], Xianghong Hu [1,*], Qiming Zhang [3] and Xiaoming Xiong [1,*]**

1   School of Automation, Guangdong University of Technology, Guangzhou 510006, China;
    hongmin_huang@163.com (H.H.); liuzihao_hello@163.com (Z.L.); taosheng_chen@163.com (T.C.)
2   Company of Chipeye Microelectronics Foshan Ltd., Foshan 528200, China
3   ZhuHai JieLi Technology Co., Ltd., Zhuhai 519015, China; zhangqiming@zh-jieli.com
*   Correspondence: xianghonghu@gdut.edu.cn (X.H.); xmxiong@gdut.edu.cn (X.X.)

**Abstract:** The You Only Look Once (YOLO) neural network has great advantages and extensive applications in computer vision. The convolutional layers are the most important part of the neural network and take up most of the computation time. Improving the efficiency of the convolution operations can greatly increase the speed of the neural network. Field programmable gate arrays (FPGAs) have been widely used in accelerators for convolutional neural networks (CNNs) thanks to their configurability and parallel computing. This paper proposes a design space exploration for the YOLO neural network based on FPGA. A data block transmission strategy is proposed and a multiply and accumulate (MAC) design, which consists of two $14 \times 14$ processing element (PE) matrices, is designed. The PE matrices are configurable for different CNNs according to the given required functions. In order to take full advantage of the limited logical resources and the memory bandwidth on the given FPGA device and to simultaneously achieve the best performance, an improved roofline model is used to evaluate the hardware design to balance the computing throughput and the memory bandwidth requirement. The accelerator achieves 41.99 giga operations per second (GOPS) and consumes 7.50 W running at the frequency of 100 MHz on the Xilinx ZC706 board.

**Keywords:** YOLO; design space exploration; PE matrices; FPGA accelerator

## 1. Introduction

Currently, convolutional neural networks (CNNs) [1] are widely applied in a great variety of fields, such as object recognition [2–5], speech recognition [6,7], facial recognition [8,9], image recognition [10–14], and so on. The acceleration of CNNs has been a popular topic of research and is implemented on small embedded devices. Since CNNs have highly parallel workloads, it is difficult for the traditional implementation platforms to achieve both a high performance and low power consumption. Due to a large number of computations of CNNs and the serial operation of the central processing unit (CPU), the CPU cannot fully exploit the parallel operation of CNNs. Recently, several CNN accelerators have been achieved in the graphics processor unit (GPU) [15]. Although GPU has efficient parallelism and a high-density computing capability [16,17], it is limited by lower energy-efficiency gain and unable to adjust the hardware resources according to various applications. The GPU is also too large to be implemented on small embedded devices. Furthermore, although displaying a high energy-efficiency gain, the application specific integrated circuit (ASIC) has a long development cycle and high cost [18,19], and it is also not cost-effective without a high volume demand. Field programmable gate arrays (FPGAs), which exhibit high parallel computing and can be programmed according to specific applications, are currently widely applied to the field of hardware

acceleration [12,20–24]. Various CNN accelerators based on FPGAs have been proposed and have become a research hotspot in the industry [25,26].

There are many studies on accelerating CNNs on embedded devices [27,28]. An analytical design scheme using the roofline model for the CNN is proposed in [20], which uses loop tiling and transformation to optimize CNN's convolution computing and reduce memory access. However, the pooling operations are not discussed in this study. Prior work [21], which takes Tiny-You Only Look Once (YOLO) as the target network, designed 64 processing elements (PEs) to exploit the computational parallelism and employed data reuse and data sharing to reduce the memory footprint. To address the parallel image processing challenge, the authors of [22] designed a new multistage architecture composed of 36 PEs and it was implemented on FPGA. A processing unit similar to the additive tree structure was used in [23]. Adopting the data reuse and task mapping techniques, the authors of [24] designed an FPGA accelerator based on OpenCL, which was able to achieve a peak performance of 33.9 giga operations per second (GOPS). Although many different accelerators have been implemented on the FPGA platforms, there are still many improvements that need to be made in terms of the performance.

The YOLO neural network, which is a kind of CNN, is able to predict the positions and categories of multiple boxes at once [2]. Because it can directly predict the positions and categories of objects in images with only one glance, YOLO can realize end-to-end object detection and recognition. Moreover, YOLO takes the object detection as a regression problem, so it has a rapid recognition speed. At present, YOLO has been developed to the fifth generation (YOLO v5). However, YOLO v2-tiny [29], as the second-generation lightweight version of YOLO, has the advantages of a fast speed, smaller memory footprint and so on. In this work, we select YOLO v2-tiny as the recognition algorithm. In order to achieve a high performance, several measures are proposed in the accelerator. After analyzing the network structure of YOLO v2-tiny, we propose a data block transmission method for YOLO v2-tiny on FPGA. In order to make full use of the hardware resources on the FPGA, we design an accelerator with two $14 \times 14$ PE matrices, which can compute in parallel, to accelerate the operations of the neural network. At the same time, in order to ensure that the data transmission can meet the computational requirements, we reduce the complexity of the design and it can be configured for other CNNs.

In this paper, the main contributions are as follows:

(1)　We analyze the structure of the YOLO v2-tiny, present a data block transmission method to transmit the feature maps efficiently, and apply the output data reuse pattern to reduce the data access to off-chip memory;
(2)　We design an accelerator architecture for YOLO v2-tiny that comprises two $14 \times 14$ PE matrices which work in parallel to achieve a high performance;
(3)　We employ the roofline model to explore all of the design space of the accelerator to identify the best performance when employing the limited hardware resources. The experimental results indicate that our architecture is better than others in terms of the metrics of the performance and the power efficiency.

The structure of the paper is organized as follows. Section 2 briefly introduces the structure and various operations of YOLO v2-tiny. Section 3 presents the architecture of the accelerator for YOLO v2-tiny and introduces the implementation of the accelerator in detail. Section 4 describes the experimental setup and a comparison with other previous works. Finally, Section 5 summarizes this paper.

## 2. YOLO v2-Tiny Neural Network

In this section, we describe the structure of YOLO v2-tiny and the computations of each layer.

As shown in Table 1, YOLO v2-tiny consists of 15 layers and is able to recognize 80 objects based on the Common Objects in Context (COCO) dataset [20], and the max pooling layers always follow the convolutional layers. In YOLO v2-tiny, the sizes of the convolution kernel are $3 \times 3$ and $1 \times 1$ with the

stride $S = 1$, and the size of the pooling kernel is $2 \times 2$ with the stride $S = 1$ or 2. The number of channels increases with an increase in the number of layers, but decreases at the last layer. Moreover, the size of the input feature maps is $416 \times 416 \times 3$ and the output size of the CNN becomes $13 \times 13 \times 425$. The 32-times reduction in the feature maps is mainly accomplished by the pooling layer.

**Table 1.** Structure of You Only Look Once (YOLO) v2-tiny.

| Layer | Type | Kernel Size ($k$), Stride ($S$) | Input Size ($H \times L$), Input Channels ($N$) | Output Size ($R \times C$), Output Channels ($M$) |
|---|---|---|---|---|
| 1 | Conv | $3 \times 3, 1$ | $418 \times 418, 3$ | $416 \times 416, 16$ |
| 2 | Max Pool | $2 \times 2, 2$ | $416 \times 416, 16$ | $208 \times 208, 16$ |
| 3 | Conv | $3 \times 3, 1$ | $210 \times 210, 16$ | $208 \times 208, 32$ |
| 4 | Max Pool | $2 \times 2, 2$ | $208 \times 208, 32$ | $104 \times 104, 32$ |
| 5 | Conv | $3 \times 3, 1$ | $106 \times 106, 32$ | $104 \times 104, 64$ |
| 6 | Max Pool | $2 \times 2, 2$ | $104 \times 104, 64$ | $52 \times 52, 64$ |
| 7 | Conv | $3 \times 3, 1$ | $54 \times 54, 64$ | $52 \times 52, 128$ |
| 8 | Max Pool | $2 \times 2, 2$ | $52 \times 52, 128$ | $26 \times 26, 128$ |
| 9 | Conv | $3 \times 3, 1$ | $28 \times 28, 128$ | $26 \times 26, 256$ |
| 10 | Max Pool | $2 \times 2, 2$ | $26 \times 26, 256$ | $13 \times 13, 256$ |
| 11 | Conv | $3 \times 3, 1$ | $15 \times 15, 256$ | $13 \times 13, 512$ |
| 12 | Max Pool | $2 \times 2, 1$ | $13 \times 13, 512$ | $13 \times 13, 512$ |
| 13 | Conv | $3 \times 3, 1$ | $15 \times 15, 512$ | $13 \times 13, 1024$ |
| 14 | Conv | $3 \times 3, 1$ | $15 \times 15, 1024$ | $13 \times 13, 512$ |
| 15 | Conv | $1 \times 1, 1$ | $13 \times 13, 512$ | $13 \times 13, 425$ |

The basic flow of YOLO v2-tiny is as follows:

(1)　The size of the external input images is adjusted to $416 \times 416$, and they are padded with 0 and turned into $418 \times 418$ as the input feature maps of the CNN;

(2)　Then, the probability and the location of the objects in the image are predicted after running YOLO v2-tiny;

(3)　Finally, the optimal results are selected by non-maximum suppression [30] in the region layer and marked on the image.

YOLO v2-tiny is composed of convolutional, batch normalization, pooling, activation function, and region layers, and so on. According to our knowledge, the convolutional layers take up over 90% of the running time [31]. Therefore, this work will mainly introduce the acceleration of convolution and pooling operations; it should be noted that although the region layer is implemented on the software, it is not introduced here.

*2.1. Convolution Structure*

The convolutional layers are used to extract different features in the images, as shown in Figure 1. Before performing the convolution operations, we need to fill the edges of each input feature map with 0 in the $3 \times 3$ convolutional layer to increase its length and width by 2. The size of input feature maps is $H \times L$ and the number of input channels is $N$. There are $M$ convolution kernels whose sizes are $k \times k$ and the number of channels is $N$. After the convolution operations, the size of the output feature maps is $R \times C$ and the number of output channels is $M$. As shown in Equations (1) and (2), we can obtain the height $R$ and width $C$ of output feature maps. In convolution operations, each convolution kernel is slid at the stride $S$, and at the same time, the pixels from each sliding window are multiplied by the corresponding weights from the convolution kernels. Finally, as shown in Equation (3), the sums of multiplications plus the biases become the pixels of the output feature maps, which are arranged sequentially according to the output channel $M$.

$$R = \frac{H - k}{S} + 1, \tag{1}$$

$$C = \frac{L-k}{S} + 1. \tag{2}$$

The convolution operations can be described as follows:

$$x = \sum_{i}^{n}(X^i \times W^i) + b, \tag{3}$$

where $x$ represents the pixel after convolution, $X$ stands for the input pixel, $W$ represents the weight, $b$ represents the bias, and $n = k \times k \times N$ stands for the number of input pixels for convolution operations.

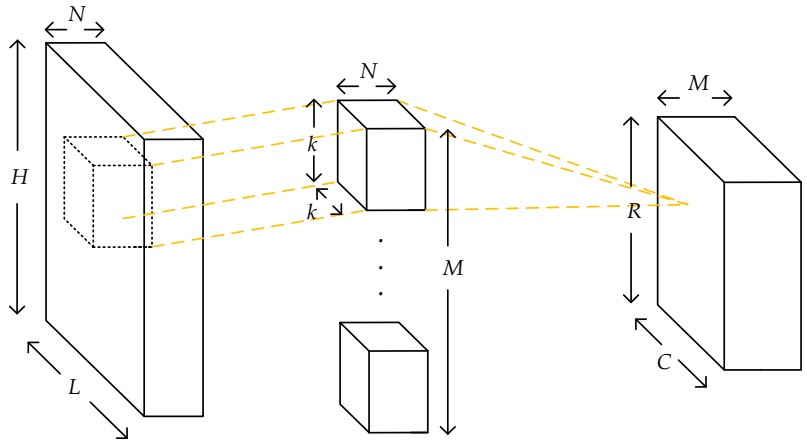

**Figure 1.** Convolution operation.

## 2.2. Batch Normalization

In order to avoid over-fitting, YOLO v2-tiny adds batch normalization [32] between the convolutional layer and the activation function. In addition, the batch normalization can improve the speed of training, speed up convergence, reduce the network's insensitivity to the initialization of weights, and so on. Given the mean $\mu$, the variance $\sigma^2$, the output pixel $y$, the scale factor $\gamma$, the offset factor $\beta$, the minimal positive number $\varepsilon$, and the $x$ from Equation (3), the function of batch normalization can be described as follows:

$$\mu = \frac{1}{m}\sum_{i}^{m}x_i, \tag{4}$$

$$\sigma^2 = \frac{1}{m}\sum_{i}^{m}(x_i - \mu), \tag{5}$$

$$y = \gamma\,\frac{x-\mu}{\sqrt{\sigma^2+\epsilon}} + \beta = \frac{\gamma}{\sqrt{\sigma^2+\epsilon}}x - \frac{\gamma\mu}{\sqrt{\sigma^2+\epsilon}} + \beta. \tag{6}$$

In order to simplify the calculation, Equation (6) is transformed into Equation (7), which only has one multiplication operation and one addition operation, by replacing $\frac{\gamma}{\sqrt{\sigma^2+\epsilon}}$ and $\beta - \frac{\gamma\mu}{\sqrt{\sigma^2+\epsilon}}$ with the parameters $a$ and $c$, respectively.

$$y = ax + c. \tag{7}$$

## 2.3. Activation Function

In CNNs, the results after convolution or batch normalization, such as $y$ from Equation (7), are activated and then transferred to the next layer. If without the activation function, the output data of each layer would be the result of a linear function of the previous layer, and no matter how many layers, the expression of the whole CNN is a simple linear function that is equivalent to a single-layer network. Because it adds the nonlinear factors, the activation function can solve the problem that

the linear model cannot solve. Typically, the activation functions include the sigmoid function [33], Rectified Linear Unit (ReLU) function [34], Leaky ReLU function [35] and so on, as shown in Equations (8)–(10), respectively. However, YOLO v2-tiny only uses the Leaky ReLU function.

Sigmoid function:

$$f = \frac{1}{1 + e^{-y}}. \tag{8}$$

ReLU function:

$$f = max(0, y). \tag{9}$$

Leaky ReLU function:

$$f = \begin{cases} y, & y \geq 0 \\ 0.1y, & y < 0 \end{cases}, \tag{10}$$

where $y$ is from Equation (7).

### 2.4. Pooling Structure

Since a tremendous amount of data in the CNN will lead to over-fitting, pooling is mainly employed in reducing the dimension of feature maps and compressing the data and parameters, thus eliminating over-fitting. There are two kinds of pooling operations: Max pooling and mean pooling. YOLO v2-tiny only requires max pooling. As shown in Figure 2, the left side is $2 \times 2$ max pooling with $S = 2$ and the right side is $2 \times 2$ mean pooling with $S = 2$, which compress the input feature map from $4 \times 4$ to $2 \times 2$, thus reducing the computations of the CNN. The different colors represent different positions of $2 \times 2$ pooling kernel.

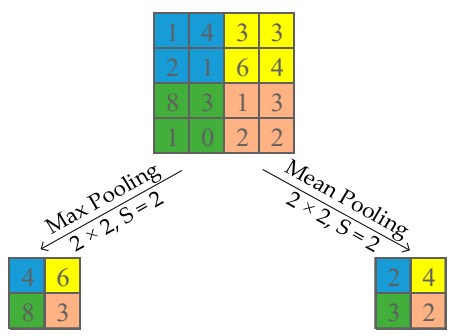

**Figure 2.** Pooling operation.

## 3. Accelerator Architecture

In this section, we introduce the architecture of the accelerator for the CNN. Firstly, we present a data block transmission strategy for YOLO v2-tiny and then introduce the structure of the accelerator with two $14 \times 14$ PE matrices. Finally, we employ the roofline model to attain the best performance of the accelerator architecture.

### 3.1. Data Transmission

Because of the limited on-chip resources of FPGA, which means that we cannot store all the data of YOLO v2-tiny at once, we need to store the data in the form of blocks to reduce the pressure of the on-chip memory. As is known, the sizes of width and height of the feature maps are times of 13 for YOLO v2-tiny, and the size of the pooling kernels is $2 \times 2$ with the stride $S = 1$ or $S = 2$. As shown in Figure 3, in order to improve the utilization of PEs, the size of the PE matrices in multiply and accumulate (MAC) is designed be to $14 \times 14$, as are the Weight Registers (WRs) and Bias Registers (BRs). In order to obtain the output feature map of $14 \times 14$, the size of the input feature map is set to

$16 \times 16$, according to the Equations (1) and (2) with $k = 3$ and $S = 1$. As shown in Figure 4, we need to cut the input feature maps into small blocks of $16 \times 16$, which will be transmitted to the input registers 1 (IR1) matrix sequentially. The front two columns of data in the latter block overlap and the upper two rows of data in the bottom block also overlap. Considering that most of the layers have lots of channels, we should divide the number of channels to reduce the pressure of the on-chip memory. Given the $Th \times Tl \times Tn$ input data block, we can attain the $Tr \times Tc \times Tm$ output data after convolution operations.

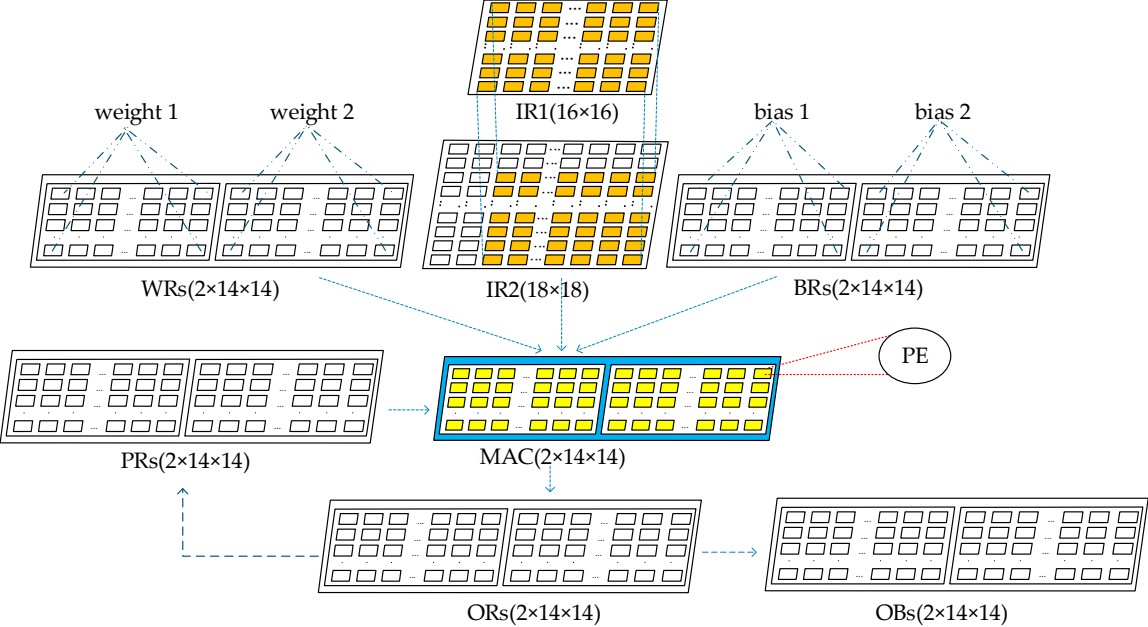

**Figure 3.** Data transmission strategy of the accelerator.

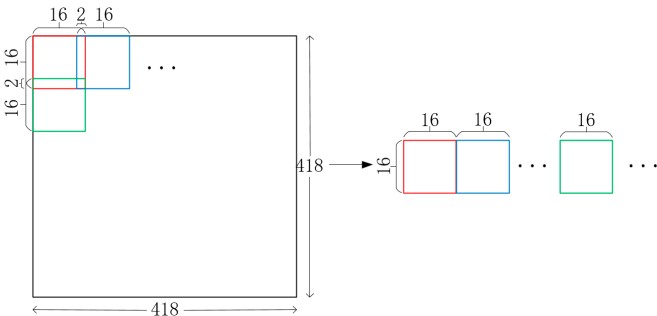

**Figure 4.** Data block transmission of $3 \times 3$ convolution, $S = 1$.

As shown in Figure 3, an accelerator consists of registers (IRs, WRs, BRs, and partial-sum registers (PRs)), MAC, off-chip memories (OBs), and interconnections between the on-chip and off-chip memory. The data transmission is divided into five layers in the accelerator, and the first layer is the IR1, which caches the input feature map of $16 \times 16$ in Figure 4 and then transmits it to IR2. The second layer consists of the IR2, the WRs, and the BRs. The pixels are shifted in IR2 and then mapped to the MAC comprised of two PE matrices. At the same time, the weights in WRs and the biases in BRs are also mapped to the corresponding PEs in MAC to perform convolution operations. IR2 also shifts pixels during the pooling operation. IR1 and IR2 operate in ping-pang mode to improve the transmission efficiency. The third layer is MAC and the fourth layer is comprised of the output registers (ORs), which transmit the intermediate data to partial-sum registers (PRs) or the final data to output buffers (OBs). The fifth layer consists of the PRs and the off-chip memories with OBs. As shown in Equation (3),

the intermediate data in PRs are transferred back to MAC to add the results of the later block, which can reduce the off-chip memory access and improve the data transmission efficiency. Unlike the hybrid data reuse patterns adopted in [36], this work only adopts the output data reuse pattern to reduce the design complexity, while maintaining a high data transmission efficiency.

Because the largest block of the input feature maps is $16 \times 16$ in YOLO v2-tiny, IR1 is set to $16 \times 16$. To realize the shifting operation of the convolution, the size of IR2 should be set to $18 \times 18$ because of kernel size $k = 3$ and stride $S = 1$. Furthermore, this design can be configured to implement other convolution operations with kernels size $k = 3/5/7$ and stride $S = 1$, and the sizes of IR1 and IR2 should be adjusted accordingly.

### 3.2. PE Structure

In this work, the image data and biases adopt 32-bit fixed points, but the weights and coefficients of Leaky ReLU use 16-bit fixed points. There are two $14 \times 14$ PE matrices in MAC, which can perform 392 multiplications and additions, or comparison operations in parallel in a single clock cycle. As shown in Figure 5, a PE consists of a multiplier, an adder, several selectors, registers and so on. The port "32_bit" is a 32-bit fixed point for image data, biases, and intermediate data. The port "16_bit" is a 16-bit fixed point for weights, coefficients of Leaky ReLU, and so on. "X0" and "X1" are the input ports for the pooling operations. The "carry" is also used for the max pooling because the subtraction is equal to reverse plus one (the complement). Furthermore, the port "partial_sum" is used to transmit the intermediate data and the "out_data" outputs the results. In order to improve the utilization of PEs, all of the computations are conducted in the PEs, which can be configured to perform various functions, such as multiplication, addition, and subtraction operations.

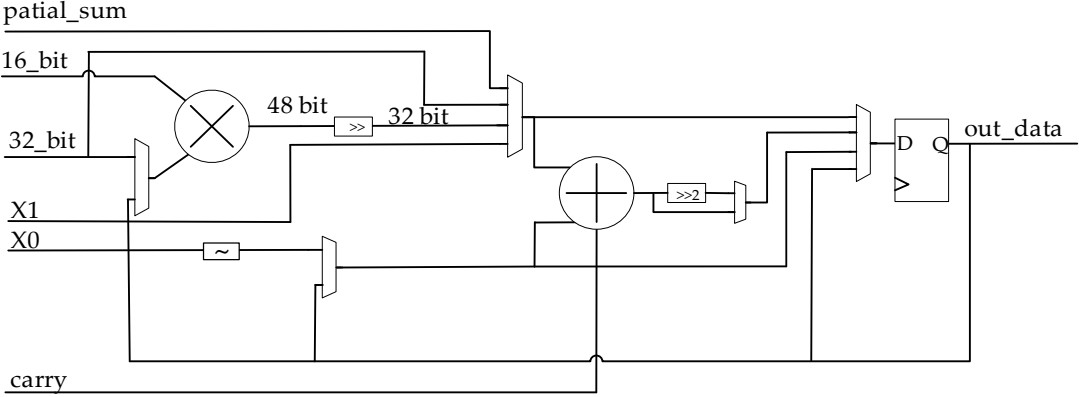

**Figure 5.** Structure of the processing element (PE).

The working frequency of the PEs is designed to be 100 MHz. The multiplier is implemented by the digital signal processors (DSPs), while the adder adopts look-up-tables (LUTs). Moreover, a 16-bit $\times$ 32-bit multiplier occupies two DSPs, of which the result is 48 bits and needs to be converted into 32 bits. The PE can realize all operations, such as convolution operation, batch normalization, activation function, and pooling operation.

### 3.3. Computational Model

Since the batch normalization always follows the convolution operations in YOLO v2-tiny, the batch normalization can be combined with the convolution operation to reduce the computational complexity. For example, Equations (3) and (7) are combined and simplified to Equation (11):

$$y = a\left(\sum_{i}^{n}(X^i \times W^i) + b\right) + c = a\left(\sum_{i}^{n}(X^i \times W^i) + b + \frac{c}{a}\right), \tag{11}$$

where the $b + \frac{c}{a}$ is a constant which can be computed in advance and cached in the BRs. Therefore, the convolution and batch normalization are combined into the convolutional layer. We will now introduce the implementation of the convolutional layer, the activation function, and the pooling layer in detail, respectively.

### 3.3.1. Convolutional Layer

In order to realize the $3 \times 3$ convolution with the stride $S = 1$ in YOLO v2-tiny, the pixels transmitted from IR1 are shifted in IR2. Figure 6 shows the mechanism of sliding windows for $3 \times 3$ convolution with $S = 1$ in IR2 (the dotted box). The pixels from $14 \times 14$ fixed positions, represented by the blue boxes in the middle of IR2, will be mapped to each PE matrix. As shown in Figure 3, because the $14 \times 14$ pixels will multiply by the same weight in each PE matrix, each WR will copy one weight into $14 \times 14$, such as "weight1" and "weight2", which are from different convolution kernels. As with WRs, each BR will copy one bias into $14 \times 14$, such as "bias1" and "bias2", which are from different output channels. At the first clock, $14 \times 14$ pixels on the blue boxes at the upper-left of IR1 (the largest solid box) are mapped to PE matrices. At the second clock, all data from IR1 are shifted one position to the left in IR2, as at the third clock. However, at the fourth clock, all of the data are moved up by one position. At the fifth and sixth clock, all data from IR1 are shifted one position to the right in IR2. Because the data at the blue box are replaced by the next data after being shifted, a PE requires nine clock cycles to receive $3 \times 3$ pixels for the $3 \times 3$ convolution. Therefore, the shifting operations in IR2 form an S-shape, such as left → left → up → right → right → up → left → left. When shifted in IR2, the pixels at the blue boxes are mapped to PE matrices, while the corresponding weights in the WRs and the corresponding biases in the BRs are also mapped to PE matrices to perform the convolution operations.

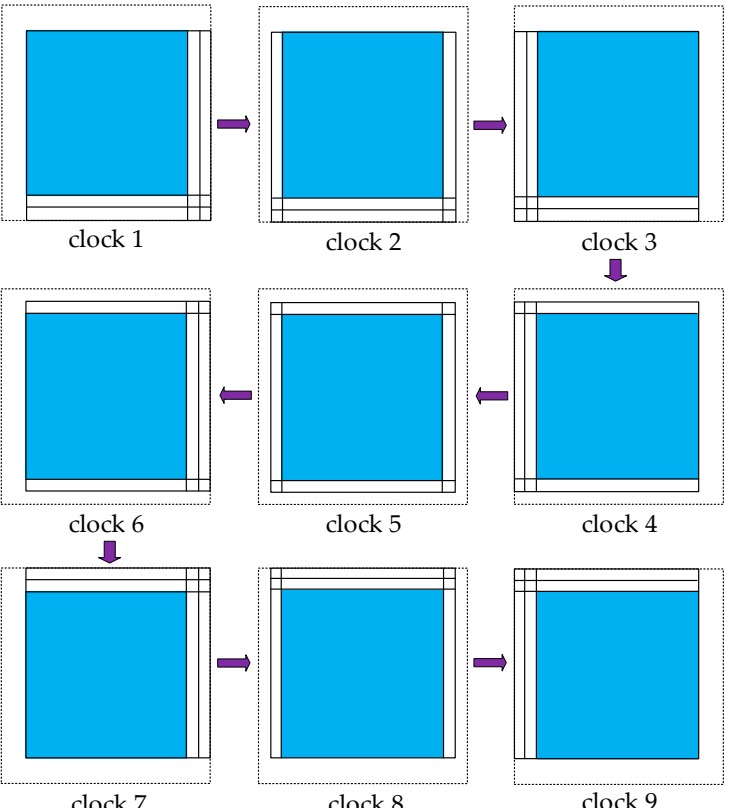

**Figure 6.** Sliding windows of input registers 2 (IR2) for $3 \times 3$ convolution, $S = 1$. Clock 1–9 represent the first to ninth clock cycles, respectively.

As shown in Equation (11), $14 \times 14$ pixels at the blue box and the corresponding weights *W* from the WRs are mapped to each PE matrix to perform the multiply-accumulate operations. The result after multiply-accumulate operations is added with the bias $b + \frac{c}{a}$ from BR. Finally, the result of the addition is multiplied by the constant *a* and the calculation of Equation (11) is completed. The data transmission path of the convolutional layer is shown in Figure 7. A PE matrix can perform $14 \times 14$ multiplication or addition operations in a single clock cycle. In this work, two PE matrices are adopted in MAC, which can achieve twice the computational efficiency.

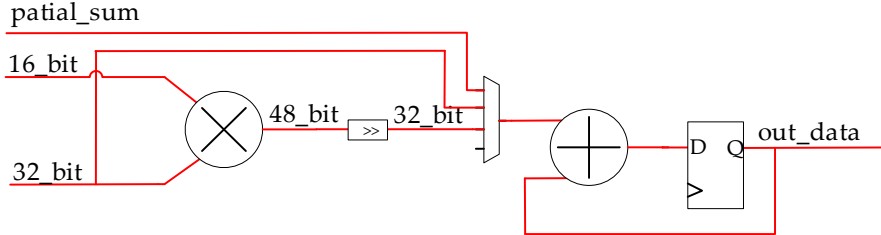

**Figure 7.** Data transmission of the convolution operation in PE.

### 3.3.2. Activation Function

The activation function used in the YOLO v2-tiny network is the Leakey ReLU function, which is connected to the convolutional layer and activates the results of the convolutional layer. After finishing Equation (11), the activation function operation in Equation (10) only uses a selector and a multiplier. When the result of the convolutional layer is positive, it is output directly; when the result is negative, it is multiplied by a coefficient of 0.1, which is 3277 represented by a fixed point. In this work, the PE can also be configured to perform other activation functions for other CNNs, such as the sigmoid function and the ReLU function.

### 3.3.3. Pooling Layer

The activation function is followed by the pooling operation. During the $2 \times 2$ max pooling operation, at the first clock cycle, each PE compares the left and right data of the $2 \times 2$ pooling kernel, and the maximum one is selected and updates the left value, which is finished in IR2. At the second clock cycle, each PE compares the above and below data, and finally the maximum of the four data is selected and outputted to the off-chip memory. As shown in Figure 8, if the result of X1-X0 is positive, X1 is greater than X0; otherwise, X0 is bigger. The principle of the mean pooling operation is the same as that of the max pooling operation. In the $2 \times 2$ mean pooling operation, the sum of the four data is divided by 4, that is, the sum is shifted by 2 bits to the right. To reduce the data transmission, the pooling operation is followed by the convolution operation, and there is no need to send the result of the convolution operation to the off-chip memory OBs.

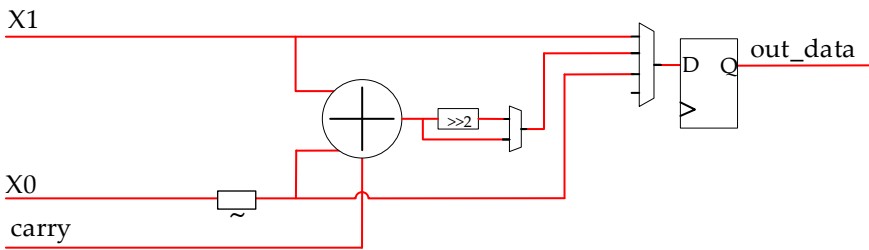

**Figure 8.** Data transmission of the pooling operation in PE.

In two clock cycles, a PE matrix can perform pooling operations at most of the 49 $2 \times 2$ pooling kernels with $S = 2$ or 169 $2 \times 2$ pooling kernels with $S = 1$. In the 12th layer Max Pool presented in

Table 1, since the sizes of input feature maps and output feature maps are $13 \times 13$, and the size of the pooling kernel is $2 \times 2$ with the pooling stride $S = 1$, it is necessary to copy the last column and last row data of the input feature maps to make them $14 \times 14$.

### 3.4. Design Space Exploration

In order to make full use of the limited FPGA resources, the well-known roofline model is adopted to explore the design space of the accelerator to achieve the best performance. As shown in Figure 9 [37], the X-axis is the operation intensity, indicating the operations of transferring to and from off-chip memory (opt/byte), and the X_max represents the maximum operation intensity when the performance is limited by the bandwidth. The peak perf. represents the performance limited by the hardware computing resources in theory. The slope $\alpha$ represents the maximum bandwidth supported by the hardware. When the operation intensity is (0, X_max), the performance is limited by the bandwidth; otherwise, it is limited by the computing resources. The best performance is the minimum of two formulas and can be calculated by Equation (12). The peak performance and the operation intensity are shown in Equations (13) and (14), respectively.

$$Performance = min \begin{cases} Operation\ intensity \times Bandwidth \\ Peak\ Performance \end{cases}, \tag{12}$$

$$Peak\ Performance = \frac{Total\ number\ of\ operations}{Total\ amount\ of\ execution\ cycles} = \frac{2 \times R \times C \times M \times N \times k \times k}{\frac{M}{Tm} \times \frac{N}{Tn} \times \frac{R}{Tr} \times \frac{C}{Tc} \times (Tm \times Tn \times k \times k + T)}, \tag{13}$$

$$Operation\ intensity = \frac{Total\ number\ of\ operations}{Total\ amount\ of\ external\ data\ access} = \frac{2 \times R \times C \times M \times N \times k \times k}{T_{in} \times DS_{in} + T_{weight} \times DS_{weight} + T_{out} \times DS_{out}}, \tag{14}$$

where $T$ represents the time of data transmission. In each convolutional layer, the total operations are $2 \times R \times C \times M \times N \times k \times k$, where parameter 2 indicates the multiplication and addition operations. In addition, $T_{in}$, $T_{weight}$, and $T_{out}$ represent the trip counts of input feature maps, weights, and output feature maps, respectively. $DS_{in}$, $DS_{weight}$, and $DS_{out}$ denote the data block sizes of input feature maps, weights, and output feature maps, respectively.

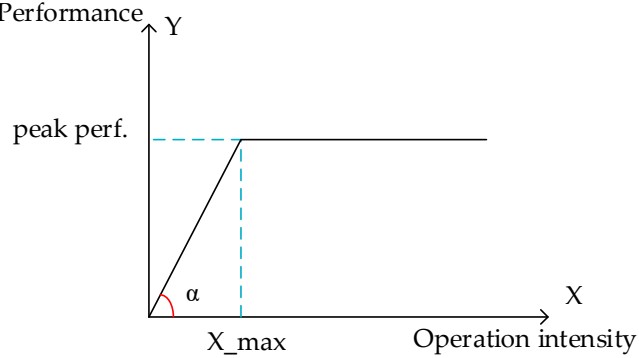

**Figure 9.** Roofline model.

Firstly, we define the size of the feature map blocks and traverse the parameters (*Tm*, *Tn*, *Tr*, and *Tc*) to determine the time for the on-chip computation. Secondly, the peak performance and the amount of off-chip memory access are calculated. Finally, through the roofline model, we can identify the best performance with the least data access.

The output data reuse pattern can reduce the output data access between the on-chip and off-chip memory, especially for the intermediate data. The intermediate data are cached in the on-chip BRAMs and then added to the next convolutional calculation block.

## 4. Experiments and Results

In this section, our scheme is employed to explore the hardware implementation on the ZC706 FPGA development board. First of all, the experimental setup of the design is introduced. Finally, the experimental results and a comparison with prior works are presented.

### 4.1. Experimental Setup

The experimental hardware platform of this work was the Xilinx ZC706 development board based on Cortex-A9 and its core was Zynq-7000 XC7Z045 FFG900C-2 SoC. This development board has 900 DSPs, many Flip Flows (FFs), LUTs, Block RAMs, and so on.

The accelerator was realized in Verilog Hardware Language. It was designed on the XilinxVivado version 2018.3 and simulated on the Mentor Modelsim version 10.5. The Vivado can synthesize and implement the Register-Transfer-Level (RTL) code to observe the resource utilization and power consumption. The Modelsim can quickly simulate the waveform to verify the function of the design. Finally, the Xilinx Software Development Kit (SDK) transplanted the YOLO v2-tiny network to the Xilinx ZC706 board for acceleration with the operating frequency of 100 MHz.

### 4.2. Experimental Results

The functional simulation of the RTL code was completed on the Modelsim and the signal simulation results of PE are shown in Figure 10.

**Figure 10.** The signal simulation results of PE.

In order to make good use of the resources of the ZC706 development board, two 14 × 14 PE matrices were designed in this design. Compared with the 16 × 16 PE matrix in [38], the 14 × 14 PE matrix reduces the design complexity and improves the utilization rate of PEs for YOLO v2-tiny. Because of the configurability, this design can realize various functions of different CNNs. In order to more accurately verify whether the design meets the resource utilization and timing, it needed to be synthesized and implemented by the Vivado to generate a bit-stream file, which was downloaded to FPGA by the software SDK. The resource utilization is shown in Table 2, where it can be seen that this design occupies 87.11% of DSP, approximately 83.29% of LUTs and 55.23% of BRAM resources.

As shown in Table 3, the adjacent convolutional layer, activation function, and pooling layer are combined into one layer, which can reduce the computation operations and data access. Moreover, the running time and performance of each layer are given. The total operations of YOLOv2-tiny are $5406.43 \times 10^6$ and the total time is 128.74 ms, so the average performance of YOLOv2-tiny is 41.99 GOPS.

**Table 2.** The overall resource utilization of the design.

| Resource | Available | Used | Utilization (%) |
|----------|-----------|------|-----------------|
| LUT  | 218,600 | 182,086 | 83.29 |
| FF   | 437,200 | 132,869 | 30.39 |
| BRAM | 545     | 301     | 55.23 |
| DSP  | 900     | 784     | 87.11 |

**Table 3.** Running time and performance per layer of YOLOv2-tiny.

| Layer | Type | Time (ms) | Operations ($10^6$) | Performance (GOPS) |
|-------|------|-----------|---------------------|--------------------|
| 1 | Conv + Max Pool | 3.48 | 149.52 | 42.97 |
| 2 | Conv + Max Pool | 8.25 | 398.72 | 48.33 |
| 3 | Conv + Max Pool | 9.39 | 398.72 | 42.46 |
| 4 | Conv + Max Pool | 9.39 | 398.72 | 42.46 |
| 5 | Conv + Max Pool | 9.39 | 398.72 | 42.46 |
| 6 | Conv + Max Pool | 14.87 | 398.72 | 26.81 |
| 7 | Conv7 | 31.33 | 1594.88 | 50.91 |
| 8 | Conv8 | 31.33 | 1594.88 | 50.91 |
| 9 | Conv9 | 11.31 | 73.55 | 6.50 |
| Total | - | 128.74 | 5406.43 | 353.81 |

Table 4 shows a comparison of our accelerator and previous accelerators. The authors of [20] presented a 32-bit floating-point design based on ImageNet, which achieves 61.62 GOPS and consumes 2240 DSPs with a power of 18.61 W. Although the design can attain a peak performance of 61.62 GOPS, it consumes a lot of DSPs. Moreover, the design in [20] only accelerates the convolutional layer, without taking into account the acceleration in the pooling layer. In Reference [39], the authors present an accelerator using the 8–16 bit fixed points based on OpenCL and its performance is low, with a value of only 30.9 GOPS. In addition, the accelerator of [39] consumes 25.8 W, which is a much higher value than that obtained with our accelerator. In Reference [24], an open-source FPGA accelerator based on OpenCL is proposed and can achieve 33.9 GOPS using only 162 DSPs. However, its precision is 32-bit float point and it has a large power of 27.3 W, so the accelerator has a low energy efficiency. Finally, the accelerator reported in [40] is designed for Yolo v2-tiny and implemented on FPGA with 16 fixed points, and its performance is 21.6 GOPS. In our experiment, we could attain the performance of 41.99 GOPS at 100 MHz, which is better than that of [24,39,40], but worse than that of [20]. In addition, the power in our design is only 7.5 W, which is far lower than that of others, and it is highly suitable for the embedded devices. The power efficiency of this design is 5.6 GOPS/W, which is better than that of others. Moreover, our accelerator has a high configurability and can be employed in several CNNs.

**Table 4.** Comparison with other works.

| Architecture | [20] | [39] | [24] | [40] | This Work |
|--------------|------|------|------|------|-----------|
| Precision | 32-bit float | 8–16-bit fixed | 32-bit float | 16-bit fixed | 16–32-bit fixed |
| Networks | ImageNet | VGG-16 | PipeCNN | Yolo v2-tiny | Yolo v2-tiny |
| Frequency | 100 MHz | 120 MHz | 181 MHz | 117 MHz | 100 MHz |
| Platform | Virtex7 VX485T | Stratix-V GXA7 | Stratix-V GXA7 | Cyclone V PCIe | Xilinx ZC706 |
| Perf. (GOPS) | 61.62 | 30.9 | 33.9 | 21.60 | 41.99 |
| DSPs | 2240 | 246 | 162 | 122 | 784 |
| Power(W) | 18.61 | 25.8 | 27.3 | - | 7.50 |
| Power efficiency (GOPS/W) | 3.31 | 1.20 | 1.24 | - | 5.6 |
| Configurability | low | low | low | low | high |

## 5. Conclusions

In this paper, we propose an accelerator for YOLO v2-tiny and implement it on the Xilinx ZC706 board with 16-/32-bit fixed points. According to the architecture of YOLO v2-tiny, a data block

transmission strategy is presented, which can reduce the pressure of the on-chip memory and improve the data transmission efficiently. The accelerator employs the output data reuse pattern to reduce the data access between the on-chip and off-chip memory, as well as the complexity of the design. In order to improve the hardware resource utilization and the computational efficiency, two $14 \times 14$ PE matrices are designed for accelerating YOLO v2-tiny. In addition, the PE matrices can be configured to realize different functions for various CNNs. Finally, in order to attain the best performance, we applied the roofline model to explore the design space with limited resources in FPGA. The results show that the accelerator achieves the performance of 41.99 GOPS at the frequency of 100 MHz and consumes 784 DSPs on the Xilinx ZC706 board. It is indicated from the experimental results that our architecture is better than others in terms of the metrics of the performance and the power efficiency. After a series of improvements, we will be able to apply the accelerator to object recognition in embedded devices, which has great practical significance. Finally, this accelerator consumes only 7.5 W, which is more suitable for small embedded devices than the GPU.

**Author Contributions:** Conceptualization, H.H. and Z.L.; methodology, H.H. and X.H.; hardware and software, H.H., Z.L. and T.C.; validation, H.H., T.C. and X.X.; investigation, H.H. and T.C.; data curation, X.H. and Q.Z.; writing—original draft preparation, H.H.; writing—review and editing, X.H. and X.X.; supervision, Z.L. and X.X.; project administration, X.X.; funding acquisition, Q.Z. and X.X. All authors have read and agreed to the published version of the manuscript.

**Funding:** This research was funded by the Key-Area Research and Development Program of Guangdong Province (No. 2019B010142001 and No. 2019B010141002).

**Acknowledgments:** The authors would like to thank the reviewers and editors for their hard work on this research.

**Conflicts of Interest:** The authors declare no conflict of interest.

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
