# Peer review of "Design Space Exploration for YOLO Neural Network Accelerator"

_electronics, doi:10.3390/electronics9111921_

Round 1

Reviewer 1 Report

The paper fits the scope of the Journal. However, its quality of presentation is low, with several English language and grammar mistakes throughout the manuscript, often preventing the reviewer to understand the overall meaning of a sentence. Several typos are also present and the references are also badly formatted.

Furthermore, I suggest including a stronger take home message in the Conclusions section, stating the usefulness and practical applications of the solution developed.

Reviewer 2 Report

Neural Networks and their implementation on FPGAs is a high interest subject in the current research work of many. The paper presents a design space exploration for YOLO neural networks using FPGAs. 

The authors clearly present the context, in a manner that not only describes the concepts, methods, results and conclusions, but also connects their current work with the work of peers in the research field.

The structure of the paper is logic, and the methods and results presented are relevant to the subject.

The presentation is done in a well written English language. 

My suggestion for the authors would be strictly related to Figure 10 (maybe enlarge it a little) and for the contribution list in Section 1 (maintain the same style - Gerund or Present Simple - "we analyze, we design, we employ" or "analyzing, designing, employing").

Reviewer 3 Report

The paper discusses acceleration architecture of YOLO convolutional neural network used for computer vision applications on FPGA. Proposed optimizations focus on the convolution layer of the algorithm and aim at reducing the data access to off-chip memory and at achieving highest performance within limited resources.

The design space exploration for YOLO v2-tiny neural network and the different options are well presented and discussed in detail. The main contributions of the proposed architecture include a) a data block transmission method to transmit the feature maps and exploit data reuse and b) matrix architecture for parallel processing. The roofline model is used to explore the design space to achieve best performance under hardware resources constraints.

The proposed architecture has been evaluated on Xilinx ZC706 308 FPGA development board with respect to performance, resource utilization and power. The proposed implementation is compared to existing work. The proposed architecture achieves best results with respect to power efficiency but not with respect to performance and performance density.

The major issue however is the fact that the comparisons are not done on the basis of a common implementation platform. Furthermore the proposed implementation should be compared with a GPU based implementation since GPUs are also widely used for the implementation of Convolutional Neural Networks.

Round 2

Reviewer 1 Report

The paper was significantly improved with respect to the previous version. The sentences are now more clear and the flow of the experimental part reported is reasonable.

I would just suggest the check of all typos still present throughout the manuscript prior its acceptance.

Author Response

Point 1: English language and style are fine/minor spell check required

Response 1: Thank you very much for your suggestions. We have made a lot of corrections for the presentation throughout the manuscript, and re-submitted the manuscript.

Point 2: I would just suggest the check of all typos still present throughout the manuscript prior its acceptance.

Response 2: Thanks for your suggestion. We have submitted our manuscript to the English editing services provided by MDPI and we also checked all typos throughout the manuscript again.

Reviewer 3 Report

The authors have introduced appropriate modifications and/or provided adequate explanations for all issues raised during the first review.

Author Response

Thanks for your suggestion. We have submitted our manuscript to the English editing services provided by MDPI and we also checked all typos throughout the manuscript again.